# Optimised Skeletal Muscle Mass as a Key Strategy for Obesity Management

**DOI:** 10.3390/metabo15020085

**Published:** 2025-02-01

**Authors:** Thomas M. Barber, Stefan Kabisch, Andreas F. H. Pfeiffer, Martin O. Weickert

**Affiliations:** 1Warwickshire Institute for the Study of Diabetes, Endocrinology and Metabolism, University Hospitals Coventry and Warwickshire, Clifford Bridge Road, Coventry CV2 2DX, UK; t.barber@warwick.ac.uk; 2Division of Biomedical Sciences, Warwick Medical School, University of Warwick, Coventry CV1 5FB, UK; 3NIHR CRF Human Metabolism Research Unit, University Hospitals Coventry and Warwickshire, Clifford Bridge Road, Coventry CV2 2DX, UK; 4Department of Endocrinology and Metabolic Medicine, Campus Benjamin Franklin, Charité University Medicine, Hindenburgdamm 30, 12203 Berlin, Germanyandreas.pfeiffer@charite.de (A.F.H.P.); 5Deutsches Zentrum für Diabetesforschung e.V., Geschäftsstelle am Helmholtz-Zentrum München, Ingolstädter Landstraße, 85764 Neuherberg, Germany; 6Centre for Sport, Exercise and Life Sciences, Faculty of Health & Life Sciences, Coventry University, Coventry CV1 5FB, UK

**Keywords:** obesity, metabolism, skeletal muscle

## Abstract

The ‘Body Mass Index’ (BMI) is an anachronistic and outdated ratio that is used as an internationally accepted diagnostic criterion for obesity, and to prioritise, stratify, and outcome-assess its management options. On an individual level, the BMI has the potential to mislead, including inaccuracies in cardiovascular risk assessment. Furthermore, the BMI places excessive emphasis on a reduction in overall body weight (rather than optimised body composition) and contributes towards a misunderstanding of the quiddity of obesity and a dispassionate societal perspective and response to the global obesity problem. The overall objective of this review is to provide an overview of obesity that transitions away from the BMI and towards a novel vista: viewing obesity from the perspective of the skeletal muscle (SM). We resurrect the SM as a tissue hidden in plain sight and provide an overview of the key role that the SM plays in influencing metabolic health and efficiency. We discuss the complex interlinks between the SM and the adipose tissue (AT) through key myokines and adipokines, and argue that rather than two separate tissues, the SM and AT should be considered as a single entity: the ‘Adipo–Muscle Axis’. We discuss the vicious circle of sarcopenic obesity, in which aging- and obesity-related decline in SM mass contributes to a worsened metabolic status and insulin resistance, which in turn further compounds SM mass and function. We provide an overview of the approaches that can mitigate against the decline in SM mass in the context of negative energy balance, including the optimisation of dietary protein intake and resistance physical exercises, and of novel molecules in development that target the SM, which will play an important role in the future management of obesity. Finally, we argue that the Adipo–Muscle Ratio (AMR) would provide a more clinically meaningful descriptor and definition of obesity than the BMI and would help to shift our focus regarding its effective management away from merely inducing weight loss and towards optimising the AMR with proper attention to the maintenance and augmentation of SM mass and function.

## 1. Introduction

The global prevalence of obesity now exceeds 890 million, with overweight affecting around 2.5 billion adults [1]. The World Health Organisation (WHO) classifies obesity as a non-infectious global pandemic [1]. This positioning of obesity as a worldwide public health problem promotes renewed and redoubled efforts to develop and implement effective preventive, diagnostic, and management strategies. Obesity confers a substantial burden upon humanity, with impairments in quality of life and wellbeing that stem from its association with wide-reaching negative sequelae including impairments in psychosocial functioning [2] and work productivity [3]. Furthermore, obesity contributes towards a major component of global healthcare expenditure [4], originating from both the direct costs of obesity management, and the indirect costs associated with >50 obesity-related co-morbidities. Indeed, many chronic diseases that manifest within the modern-day clinical arena stem directly from weight gain and obesity or are at least pathophysiologically vulnerable to the metabolic and/or physical ramifications of obesity. Within this context, Type 2 Diabetes Mellitus (T2D) is notable, including a plethora of other obesity-related metabolic dysfunctional conditions such as dyslipidaemia, dysglycaemia, hypertension, Obstructive Sleep Apnoea (OSA), Polycystic Ovary Syndrome (PCOS), and Metabolic-Associated Fatty Liver Disease (MAFLD) [5]. Furthermore, obesity is associated with multiple malignancies that in turn contribute towards premature mortality [6,7]. Insulin resistance (IR, often associated with oxidative stress and chronic inflammatory status) is a key mediator between obesity and its adverse metabolic sequelae, and develops in the context of excessive adiposity, particularly visceral adiposity and ectopic fat deposition (epicardial, hepatic, and intramyocellular lipid [IMCL]) [8]. Obesity is a heterogeneous clinical entity with distinct subtypes based on underlying genetic architecture and clinical phenotypic biomarkers that include measures of insulin sensitivity, glycaemia, physical fitness, body composition, and cardiovascular risk [9].

One key challenge for the field of obesity is the relative lack of refinement concerning its definition. This has compounded societal misunderstanding of obesity and perhaps has also contributed towards a widespread stigmatisation and dispassionate perspective of obesity [10]. Such lack of diagnostic refinement has also stymied the development and implementation of more effective and targeted management guidelines for obesity from esteemed societies and has resulted in an unfortunate over-emphasis on ‘percentage body weight loss’ as a key outcome variable for both the clinical management of obesity and the assessment of novel therapies for obesity. The concept of ‘body weight divided by height squared’ as an indicator of body size was originally proposed by Quetelet and was widely adopted as the ‘Body Mass Index’ in the 1980s [11,12]. Since then, our definition of obesity has rested firmly on the BMI. To account for differences in overall cardiovascular risk between ethnic-specific groups based on the BMI, NICE provide ethnic-specific cut-offs for the BMI to define obesity. However, population-based data from a cohort in England demonstrate marked ethnic-specific variance in the risk of developing T2D based on the BMI [13]. To validate and confirm these insights, there is a requirement for further similar population-based studies on cohorts from different geographical locations and with controls for potential confounders such as socio-economic groups.

Most guidelines on obesity management from esteemed societies refer solely to the BMI [14]. There are various problems with the BMI as a key solitary definition of obesity (and by implication nutritional status), not the least of which being that the BMI fails as a meaningful clinical entity at extremes of body habitus (in which there is either excessive or severely diminished muscularity) [12]. Furthermore, the BMI provides no indication of overall adiposity or body fat distribution (including the amount of visceral or ectopic fat). At one extreme, there are people with excessive muscularity (such as body builders) with lean mass exceeding 70% in some cases, in whom the BMI is elevated accordingly. Such a clinical scenario creates potential for confusion and mismanagement based on the imaginatively impoverished and unidimensional, but internationally accepted, BMI criteria for defining obesity and stratifying and prioritising its management approach. Whilst some readers may think that using the example of a body builder is meaningless as the absence of excessive adiposity is obvious clinically, there are plenty of other examples that are relatively common within the clinical arena where accurate assessment of the musculature is challenging, in which cases the application of the BMI can genuinely create confusion and mismanagement. Furthermore, the example of body builders serves to illustrate the inadequacy of the BMI as a diagnostic and management-determining criterion for obesity: it is unfit for clinical usage in the 21st century. Most frustratingly, the BMI does not provide an accurate and reliable clinical indicator of cardiovascular risk, which would provide a more logical scaffold than the BMI on which to stratify obesity-focused management options. Although commonly used for individual patients, it is interesting to note that when the BMI was first conceived by Ancel Keys in the 1970s, it was proposed for use in population-wide epidemiological settings rather than as a diagnostic tool in individual patients [15], and as such the BMI is not even a component of some composite risk scores for the estimation of cardiovascular risk such as the Framingham score or PROCAM. A re-definition of obesity would require international consensus and demand focused attention on the quiddity of obesity. Whilst ‘excessive adiposity’ would generally be regarded as an essential criterion, there are many questions that relate to age- and sex-dependent normal ranges, inter- and intra-individual differences in adiposity, and the development and implementation of clinical tools (including non-invasive biomarkers) for the accurate and reliable measurement of adiposity, including its distribution and inflammatory status. The process of such a re-definition of obesity would also likely broaden our minds beyond the role of adiposity per se in both the development of obesity but also as a sole treatment target and clinical outcome measure. Indeed, to properly re-define obesity we would need to take a more holistic metabolic perspective of adiposity in its wider pathophysiological context and consider important roles of other tissues and processes.

Within this wider context, the skeletal muscle (SM) looms as a cynosure, but it is eclipsed by its surrounding adiposity. Unfortunately, the role of the SM in the development and management of obesity has been neglected, partly because of an over-emphasis on the BMI, but also due to a greater focus on the adipose tissue (AT). Recently, however, the SM has received attention in relation to clinical data from Glucagon-like Peptide 1 (GLP1)-related pharmacotherapies for T2D and obesity. It is therefore both timely and topical to review this topic. There is a close physical proximity and interplay between the SM and AT regarding their anatomy and physiology, a scenario that is ripe for bidirectional paracrine effects. Changes within the AT influence the SM and vice versa. In this concise review, we provide an overview of the key physiological roles of the SM and discuss the multitudinous interactions between the SM and AT in the wider context of metabolic health. We discuss the association of obesity with sarcopenia (a form of SM atrophy), the metabolic sequelae of sarcopenia (including the key role of the SM in determining insulin sensitivity), and the inherent difficulties of clinical assessment of SM mass and functioning in the context of obesity. We discuss the mechanisms that contribute to the loss of SM mass during body weight loss (including in the context of pharmacotherapies for obesity) and strategies to preserve SM mass and functioning, including the recent development of pharmacotherapies that can preserve and maintain SM mass and thereby mitigate against the loss of SM mass during weight loss, and their likely positioning in future treatment algorithms for obesity. Finally, we outline a future scenario in which body composition (the ‘Adipo–Muscle Ratio [AMR]’) rather than the BMI assumes a central position as a re-defined diagnostic criterion for obesity, and in which key clinical outcome measures focus on changes in body composition (including the AMR) rather than the percentage of body weight loss per se.

## 2. Methods

We performed a narrative review of the current literature, using Pubmed. The search terms included ‘obesity’, ‘metabolism’, and ‘skeletal muscle’ (without inclusion of any subtypes or phenotypes of obesity). We chose published studies for inclusion in our narrative review based on perceived clinical relevance, novelty, and relative size, as assessed by the authors. We included all types of peer-reviewed articles, including narrative reviews, systematic reviews and meta-analyses, short communications, clinical cases, and original articles. There were no restrictions on the date of publication, and we only considered articles written in English.

## 3. Results

### 3.1. The Physiological Role of the SM and Its Decline with Aging

There are three main types of musculature: cardiac, smooth, and skeletal. For the purposes of this review, we focus on the SM, which forms a substantial proportion of body weight and composition. In a historical study using a whole-body MRI approach in men and women (n = 468), it was shown that SM mass accounted for 38.4% and 30.6% of body mass, respectively, with gender differences greater in the upper than lower body [16]. In this study, there was an association between reductions in the SM and aging, with reductions starting in the third decade and becoming noticeable beyond the age of 50 years, primarily in the lower body [16]. Body height and weight accounted for around 50% of the variance in SM mass in both men and women, with linear and curvilinear relationships, respectively (the latter being due to the SM having less of a contribution towards weight gain with increasing body weight) [16].

The SM organ system is the largest in the body [17]. The primary physiological roles for the SM include the coordination of physical movements and locomotion, posture, breathing, and temperature homeostasis [17]. The SM consists of striated muscles that are constructed of a vast array of sarcomeres [18], with muscle fibre bundles that vary in type, including slow-twitch (type 1) and fast-twitch (type 2) [17]. The SM attaches to bones via tendons and can be viewed as a ‘force producer’ that is sensitive to both length and velocity [18]. The SM manifests as a wide variety of shapes and sizes for multiple tasks, with SM architecture and moment arms uniquely evolved for specific physical actions [18], under voluntary control by the somatic nervous system [17]. In addition to its key role in movement and locomotion, the SM also provides physical protection from traumatic injury and soft tissue support of the internal organs [17]. Furthermore, optimised SM strength reduces vulnerability from falls and other injuries that can result in fracture (notably hip fracture), which in turn can impact on co-morbidity, wellbeing, mortality, and healthcare expenditure.

In addition to the key physiological functions of the SM outlined above, the SM is also a major contributor to metabolic functioning and efficiency, including protein metabolism and glucose homeostasis [17]. A secondary function of the SM is that it acts as a reservoir for amino acids (AAs) that are released when additional protein supplies are needed, including in the context of disease with enhanced metabolic requirements and a diminished intake of dietary protein [19]. Furthermore, the SM accounts for >80% of glucose uptake in the post-prandial period following an oral glucose load, with linear increases in uptake over time [17,20]. It follows that IR in the SM plays a major role in the development of hyperglycaemia and T2D and other metabolic dysfunctions in the context of obesity, with a diminished overall uptake of glucose by the SM [17]. Indeed, IR within the SM can develop independently of and decades prior to the onset of β-cell failure and the onset of T2D in those at high metabolic risk [17,21]. Given its important role in glucose handling, the SM can be considered as a primary driver of whole-body IR [17]. Furthermore, unlike β-cell failure in T2D, IR within the SM is reversible and therefore represents a clinically meaningful therapeutic target to optimise insulin sensitivity and overall metabolic efficiency and flexibility, especially in the context of obesity. In addition to the SM, there is also a strong interplay between the AT and IR [22]. Indeed, with a weight loss of >10 Kg (and attendant loss of AT mass), there is recovery of β-cell function, and remission to a non-diabetic state in the context of T2D [22,23].

Beyond protein metabolism and glucose homeostasis, the SM also represents the most important determinant of the basal metabolic rate (BMR) as the largest thermogenic organ [24]. Within the SM, thermogenesis (defined as any metabolic process that dissipates energy in the form of heat) occurs through both shivering and non-shivering thermogenesis (NST) via ATP hydrolysis, in response to cold [24]. During cold acclimation, NST gradually replaces shivering. Thermogenesis also occurs within brown AT, driven by uncoupling protein 1 (UCP1), which reduces the proton gradient by the electron transport pathway within the mitochondria [24]. Through uncoupling ATP generation, intracellular energy is released as heat [24,25]. Within the SM, NST also occurs but via a separate signalling pathway that implicates sarcolipin (SLN) and the ‘sarcoplasmic reticulum Ca-ATPase’ (SERCA) pump [24]. SLN is an uncoupler that results in futile cycling of the SERCA pump, enhancing ATP hydrolysis and releasing heat in the process [24]. The activity of SLN is influenced by exercise, endoplasmic reticulum (ER) stress, muscle atrophy, and the actions of various hormones that include thyroid hormones and glucocorticoids [24,26]. Uncoupling of oxidative phosphorylation within the SM via the SERCA pump can also occur through the action of uncoupling protein 3 (UCP3) [24,27]. Furthermore, increases in cytosolic Ca^2+^ activates the Ca^2+^-dependent signalling pathway and various nuclear transcription factors that promote mitochondrial biogenesis [24,28], further enhancing the potential for thermogenic energy expenditure.

As intimated earlier, there is a gradual reduction in SM mass with aging, which often becomes meaningful clinically from around the age of 50 years [17]. This gradual aging-related reduction in SM mass (trending towards a state of sarcopenia) contributes towards an overall decline in SM strength, function, and regenerative capacity, and is a key contributor to aging-related frailty with important implications for global health and economics [29,30,31,32]. Furthermore, there is also an aging-related decline in SM-mediated metabolic function that manifests with a diminishment of mitochondrial function and capacity for thermogenic energy expenditure [17,33]. This process is an important contributor to the enhanced risk for the development of T2D (and other metabolic dysfunction) in older age, which in turn is driven by aging-associated acceleration of IR within the SM [17]. In addition to ‘normal aging’, enhanced metabolic dysfunction within the SM can also be influenced by T2D itself, other chronic diseases, and even a sedentary lifestyle, mediated through a process termed ‘secondary aging’ [17,34]. Unlike normal aging, the deleterious effects of secondary aging on the SM can be delayed through engagement in healthy lifestyle behaviours such as exercise and increased physical activity [17,35]. Unfortunately, once sarcopenia becomes manifest, this can result in a vicious circle in which sarcopenia, through associated reduced physical strength and limitation of movement, begets sedentariness and vice versa [17]. In such a scenario, metabolic decline seems inevitable. Aging-induced sarcopenia (whether through normal and/or secondary aging) results in a generalised diminishment of the metabolic benefits of SM outlined above, including an overall reduction in BMR and restriction of the availability of protein and energy. Furthermore, sarcopenia is also associated with other adverse health outcomes that include delayed recovery from illness, worsened quality of life, and physical disability [19]. These adverse health outcomes of sarcopenia can be mitigated to some extent through engagement in healthy lifestyle behaviours that include physical exercise and adequate nutrition, with a focus on a high-protein diet (HPD) that can help to rebuild and maintain muscle mass and strength during the aging process [19].

### 3.2. The ‘Adipo–Muscle Axis’ and Relevance for Metabolic Health

Having reviewed the essential role that the SM plays in the regulation of overall metabolic functioning and efficiency, and the progression of metabolic dysfunction with aging-associated diminishment of SM mass, in this section we consider the ‘Adipo–Muscle Axis’ (AMA): its physiological functioning and relevance for metabolic health (Figure 1). Although traditionally the physiological functioning and pathological potential of peripheral SM and AT have been considered as separate entities, the reality is that there is a close linkage between the SM and AT [36,37,38]. This close association manifests both anatomically, with the SM often surrounded and encased by the AT, and physiologically, through extensive crosstalk between the SM and AT that implicates bidirectional endocrine, autocrine, and paracrine effects. In this context, rather than taking the traditional perspective of considering each tissue type in isolation of the other, it is more meaningful to envision the SM and AT as a single functioning unit, at least from a metabolic perspective: an ‘AMA’. To understand the AMA better, it is useful to divide our discussion into molecules that originate from the SM (myokines) and from the AT (adipokines). This field is vast and goes beyond the scope of this concise review, and there are recent reviews that are more expansive [39]. Here, we simply provide a summary and discuss a few key myokines and adipokines to illustrate their roles in mediating crosstalk between the SM and AT.

#### 3.2.1. The Role of Myokines in the AMA

Myokines (>600 peptides identified to date) influence the physiological function of the AT and other organs including bones and the brain [17]. Amongst these myokines, some promote SM hypertrophy (such as musclin, leukemia inhibitory factor [LIF], interleukin-6 [IL-6], interleukin-7 [IL-7], and interleukin-15 [IL-15]), and some inhibit muscle hypertrophy (such as myostatin) [40,41]. Physical exercise and activity can induce the secretion of myokines with anti-inflammatory effects [17,42], mediated through the effects of IL-6 and Tumour Necrosis Factor α (TNF-α) [17]. In addition to physical exercise, whole-body Electro-Myo-Stimulation (EMS, an exercise-mimetic in which a mild electrical pulse is targeted to the SM) may be considered to improve SM function, although the impact of EMS on the release of myokines and AT functioning is incompletely understood and the effect sizes on SM function are small [43]. Therefore, the act of EMS-induced SM contraction may be insufficient to elicit myokine release with associated metabolic benefits. By implication, for physiological myokine release from the SM there is a requirement for other physiological processes that occur during physical exercise beyond mere SM contraction. In this sub-section, we discuss the effects of myokines (their release being down- or upregulated from the SM in response to physical exercise) on the metabolic functioning of the AT, illustrated here by the following three key examples:

Myostatin has received attention recently due to its therapeutic potential for preserving and enhancing SM mass. Myostatin is a highly conserved member of the transforming growth factor-β (TGFβ) family of proteins and is expressed predominantly in the SM, but also in the AT [17]. Myostatin acts as a negative regulator of SM mass, and its ablation or mutation can lead to muscle hypertrophy [17]. Following resistance exercise, there is a reduction in SM myostatin protein expression and abundance [17,44]. Interestingly, EMS also decreased serum levels of myostatin and helped to maintain SM mass in a human-based study on patients who had undergone living-donor liver transplantation [45]. The inhibition of myostatin results in enhanced SM growth due to the inhibitory effects of myostatin on myogenesis and protein growth [17]. Myostatin also mediates some of the crosstalk between the SM and AT. As a key ligand for the TGFβ family, myostatin can modulate TGFβ signalling mediated through activins and inhibins, and thereby influence the beige and brown characteristics of the AT [46].

Irisin is a fragment of ‘Fibronectin type III domain-containing protein 5′ (FNDC5) and there is some controversy in the literature regarding its measurement, which has been considered akin to ‘chasing shadows’ [47,48]. Irisin is stimulated by ‘Peroxisome proliferator-activated receptor-Gamma Coactivator 1α’ (PGC1α), and mediates beneficial metabolic effects through the beiging of white AT [17,49], thereby activating UCP1 and inducing enhanced thermogenesis within the white AT. In a combined human- and rodent-based study that explored the cerebral benefits of EMS, there was no evidence for any increased levels of irisin within the serum or SM, but rather lactate was highlighted as a mediator of the SM–brain crosstalk [50]. Conversely, in human- and rodent-based models of physical exercise, irisin transcript levels increased and this was associated with an increase in mitochondrial number and oxygen consumption [17]. The importance of irisin for metabolic effects within the SM (including enhanced insulin sensitivity and glucose uptake) in response to running was demonstrated in a mutant mouse model [17,51]. The expression and release of irisin from the SM can be downregulated by certain harmful nutrients like palmitate, via negative regulation of the expression of fibronectin type III domain-containing protein 5 (FNDC5), which in turn is another myokine that influences metabolic processes and brown AT development [17]. In this way, certain nutritional factors can influence insulin sensitivity within the SM through downregulation of insulin-sensitising factors such as irisin [17].

Interleukin-15 (IL-15) is released from the SM in response to exercise and muscle contraction, and has both anti-inflammatory and AT-reducing effects [17]. In humans, moderate-intensity physical training appears to have a positive effect on serum levels of IL-15 [17,52]. IL-15 enrichment in cultured SM can increase muscle mass [17,53]. In addition to the enhancement of fatty acid oxidation and glycogen synthesis in SM, IL-15 also has important effects within the AT, including the stimulation of a reduction in lipid accumulation and increased adiponectin secretion [17,54]. In rodent models, IL-15 has also been demonstrated to increase glucose uptake into the SM [17].

#### 3.2.2. The Role of Adipokines in the AMA

The AT releases adipokines, which have pleiotropic effects including within the SM. Unlike myokines, which are regulated primarily by physical exercise [17], adipokines are regulated by a plethora of factors that include endocrine, paracrine, and inflammatory effects, including myokines, as outlined above. Although the majority of adipokines are implicated in promoting IR within the SM stemming from excessive AT mass, around half have also been demonstrated in vitro to have effects on myogenesis and/or SM hypertrophy [55]. Importantly, there are close multidirectional links between adipokine release, adiposity, inflammatory status, and insulin sensitivity [17]. Here, we discuss the important impact of adipokines on the metabolic functioning of the SM, as illustrated by the following three key examples:

Adiponectin is released from the AT and plays a key role in regulating metabolic functioning of the SM, including enhanced mitochondrial function, Glucose Transporter 4 (GLUT4) translocation (to improve glucose uptake), and fatty acid oxidation [56]. Abdominal obesity correlates closely with reduced serum levels of adiponectin [56]. Furthermore, during moderate weight reduction in a human-based study on participants with obesity, there was a relative increase in high- and medium-molecular-weight adiponectin (and a reduction in low-molecular-weight adiponectin) [57]. Low serum levels of adiponectin are associated with poor glucose metabolism in the SM [56]. Adiponectin may also enhance myogenesis and suppress proteolysis within the SM [56]. In addition to its release from the AT, the expression of AdipoR1 (the main receptor for adiponectin) within the SM influences the effects of adiponectin. It is known that obesity, IR, and T2D are associated with a downregulation of the gene encoding AdipoR1 [56,58]. It is likely that obesity-associated downregulation of AdipoR1 expression within the SM contributes towards a diminished response of the SM to adiponectin, which in turn contributes towards IR [56].

Spexin is released from the AT and appears to facilitate glucose uptake and lipid metabolism in the SM of animals with T2D or obesity (with reduced serum levels of spexin in these conditions) [56,59]. Furthermore, exogenous administration of spexin improves insulin and glucose tolerance in rodents with diet-induced T2D and obesity [56,60]. In a further rodent-based model of IR, it was shown that spexin treatment resulted in an upregulation in the expression of genes implicated in the metabolism of glucose and lipids, including GLUT4 and PGC1α, thereby increasing insulin sensitivity within the SM [56,61]. Moreover, spexin-2 mutant zebrafish manifest IR and an increased buildup of lipids [56,62]. In summary, these animal models demonstrate metabolically beneficial effects of spexin on the SM to enhance the utilisation of glucose, thereby improving glucose homeostasis and insulin sensitivity [56]. The impact of spexin on IR in human SM is unknown. However, human obesity and T2D are associated with lower serum levels of spexin, and a lower expression of the gene encoding spexin within the AT, thereby worsening IR [56,63]. These data promote the exploration of spexin as a possible future therapeutic target for IR or even as a biomarker for metabolic dysfunction [56].

Leptin is released from the AT and has pleiotropic effects on appetite and metabolic functioning in numerous tissues, which are reviewed in detail elsewhere [64]. Leptin impacts directly on chemo-mechanical work efficiency within the SM [65]. In humans, weight loss is associated with an improved metabolic efficiency within the SM, including a decrease in the ratio of glycolytic to oxidative activity [65]. However, the extent to which leptin influences the oxidative metabolism is unclear. The field of leptin research in humans is a complicated topic. Although serum levels of leptin increase with increasing AT mass in humans, leptin resistance also increases concurrently [66]. Therefore, in the context of human obesity, although serum levels of leptin are typically elevated, there is a diminishment in the endocrine and paracrine effects of leptin due to leptin resistance at the leptin receptor, analogous to obesity-related IR. In addition to effects on metabolic efficiency within the SM, leptin also impacts on the regulation of SM mass [55]. In a rodent-based model, lipodystrophic ‘fat-free’ mice had a significantly reduced mass (15%) and peak contractile tension (20%) within fast-twitch SM compared with wild-type littermates [55]. These SM deficits were completely rescued by reconstitution of just 10% of normal AT mass, and this process was mediated solely by leptin [55]. Therefore, the normal development of SM mass and strength in rodents depends on leptin [55]. The influence of leptin on SM mass and function and other aspects of metabolic regulation (including appetite control) is complicated in humans by the association of obesity with leptin resistance [67]. The impact of leptin on the SM in human obesity remains incompletely understood.

In the context of the AMA, excessive AT can result in obesity-induced inflammation (including infiltration by immune cells such as macrophages and T-cells) within the SM, which in turn can result in the release of other ‘adipo-myokines’ from both the SM and AT [17]. Furthermore, in addition to the AT that surrounds the SM and is located subcutaneously, there is also IMCL, which can expand with obesity and account for up to 10% of total SM mass [17]. IMCL appears to have dual roles. In the context of overconsumption of dietary fat with the absence of regular physical exercise, IMCL may worsen IR within the SM [17,68]. Conversely, in the context of regular physical activity and the trained state (such as athletes), IMCL may also have a physiological role as an intracellular source of energy [68], and is associated with a higher expression of genes implicated in lipid metabolism within the SM [69]. Therefore, the AMA is a complex entity, the function of which morphs according to both the degree of adiposity and location of the AT, and additional factors such as the degree of IMCL in the context of lifestyle factors such as dietary fat intake and physical activity. Furthermore, the AMA is important for the regulation of IR, inflammatory processes, and metabolic efficiency.

### 3.3. Sarcopenic Obesity and Its Implications for Metabolic Health

Having discussed the physiological and metabolic importance of SM, its decline with age, and the complex interplay between the SM and AT through the release of myokines and adipokines, respectively, in this section we consider sarcopenic obesity (SO) and how this can impact on metabolic health. This is a complex topic that has been reviewed in detail recently [43]. Here, we provide an overview.

Unfortunately, there is no universal consensus on the diagnostic criteria for SO [43]. Sarcopenia is defined as a loss of SM function (dynapenia) and mass, and a gait speed < 0.8 m/s [43]. One challenge with the clinical assessment and diagnosis of SO is that in the context of extreme obesity the physical effort of walking and moving, coupled with the frequent association of obesity with joint and back pains, can impact negatively on gait speed and physical strength. Although body composition measures are considered clinically, these approaches are often limited by weight restrictions of imaging machines and scales and a general lack of availability and resource within healthcare settings. Adults > 65 years represent an expanding demographic globally [43]. As alluded to earlier, aging is associated with a gradual diminishment in SM mass and a progressive loss of physical strength [43,70]. In addition to aging, other factors can contribute towards the development of sarcopenia even in the context of normal body weight, including, for example, neuroendocrine tumours and other forms of malignancies. In these scenarios, the development of sarcopenia is associated with a worse survival rate [71,72].

SO is a clinical condition defined by excessive adiposity and a low SM mass and/or function [73] and affects around 11% of older adults globally, with a significant increase in prevalence beyond the age of 70 years [43,74]. As intimated earlier, although age is a strong risk factor for the development and severity of SO, this condition is not exclusive to older age, and therefore the diagnosis of SO should be based on clinical features, with age being a component of risk management [43]. There is incomplete understanding of the pathogenesis of SO. In addition to the aging-related decline in SM mass and function, there are other obesity-related factors that can diminish SM mass through the effects of adipokines and cytokines outlined in the last section. Excessive adiposity can be associated with an inflammatory response within the AT, including the release of certain pro-inflammatory cytokines like IL-6 and TNF-α, which in turn can adversely affect the SM directly and be associated with a diminished SM mass and strength [43,75]. In addition to leptin- and adiponectin-related effects on the SM, a further mechanism that contributes towards the development of SO, at least in men, is the increased aromatase expression that occurs within adipocytes in the context of excessive adiposity. Aromatase converts testosterone to estradiol, thereby contributing to the development of Male Obesity-Associated Secondary Hypogonadism (MOSH) [76]. In turn, MOSH can worsen body composition, with further diminishment of SM mass, quality, and function through impaired muscle protein synthesis and myogenic differentiation, and stimulation of adipogenesis [43,77]. In addition, the SM can impact directly on the development of SO through the depletion of mitochondria [43]. This mechanism occurs through mitochondrial fission in response to an energy burden related to obesity and aging [43]. Indeed, the restoration of oxidative phosphorylation and mitochondrial quality within the SM can rescue the regenerative failure of mitochondrial fission in aged satellite cells [43,78]. Therefore, mitochondrial function and morphology determine the function of stem cells within the SM (quiescence versus proliferation to aid tissue repair) [43], and therefore the propensity for the development of sarcopenia. Mitochondrial fitness or dysfunction underlies SM function, and it is notable that even without physical activity weight loss results in extensive changes in mitochondrial function that mediate improved metabolic efficiency of the SM [79,80]. It is beyond the scope of this concise review to explore this topic further, which is the subject of recent comprehensive reviews [81,82].

As alluded to earlier, the development of sarcopenia has serious negative consequences for metabolic health and efficiency [43]. The loss of SM mass enhances IR through a diminishment of available insulin-responsive tissue [43]. This in turn exacerbates the decline in SM mass and function through diminished anabolic signalling. To compound this scenario, the accumulation of IMCL in the context of obesity and aging accelerates the development of sarcopenia-related IR [43], and is associated with a further reduction in SM performance, strength, and mobility [43,83]. (In a rodent-based model of obesity, a reduction in IMCL content resulted in an improvement in insulin sensitivity within the SM [43,84].) The development of SO can therefore be viewed as a toxic feedback loop in which obesity and IMCL worsen IR and beget sarcopenia, and sarcopenia further worsens IR and contributes towards worsening obesity and progression of IMCL [17]. It follows therefore that any strategy for healthy aging should have at its core a clear focus on the maintenance and optimisation of SM mass and function, and mitigation against their inevitable decline with age, especially in the context of weight gain and obesity. This forms the theme for the rest of this review.

### 3.4. Mechanisms That Impact of the Loss of SM Mass with Weight Loss

Our current approaches for the management of obesity all have one final common pathway: the inducement of ‘negative energy balance’ (NEB). With lifestyle management of obesity through caloric restriction (or restriction of certain macronutrients like carbohydrates) and/or changes to dietary behaviours such as intermittent fasting, often combined with increased physical activity and exercise, weight loss occurs through NEB. Following bariatric surgical procedures like sleeve gastrectomy, endoscopic sleeve gastroplasty, and gastric bypass (and even the insertion of intragastric balloons), weight loss occurs through NEB. In this context, NEB stems from reduced food intake resulting from changes in the release of incretin hormones that suppress appetite and physical restrictions within the stomach. Furthermore, the administration of GLP1-based pharmacotherapies induces NEB through their primary effects on direct appetite suppression and, at least in the shorter-term, the slowing of gastric emptying, with a reduction in food intake.

Unfortunately, the inducement of NEB has an inevitable negative impact on SM mass. Indeed, following sustained periods of NEB, typically around 25% of total body weight loss stems from Fat-Free Mass (FFM), of which the SM forms a major component, with typically just 75% of body weight loss being from the AT [85,86]. Such a loss of SM mass has a negative impact on metabolic processes (including a downregulation of protein turnover and BMR and a worsening IR), physical performance, SM function, and susceptibility to physical injury [85]. The issue of loss of FFM in the context of voluntary weight loss is topical and of great interest currently. This renewed interest in body composition stems from a new era in GLP1-based pharmacotherapies for obesity, in which a key benchmark of 20% total body weight loss has now been surpassed, for example, in response to the dual incretin (combined GLP1 and Glucose-dependent Insulinotropic Polypeptide [GIP]) agonist, Tirzepatide, in the SURMOUNT-4 randomised clinical trial [87]. With such highly efficacious and unprecedented weight-loss pharmacotherapies, their negative impact on FFM through a much greater degree of NEB has become an important clinical concern. Indeed, in one review of studies that used Dual-Energy X-ray Absorptiometry (DXA) for measuring FFM, it was shown that FFM was reduced by between 20 and 40% of total body weight loss in response to GLP1-based therapies [88]. Although the proportion of FFM loss was highly variable, most studies included reported a FFM loss of >25% [88].

Although the mechanisms that control SM mass during NEB remain incompletely understood, AA tracer techniques have provided some useful insights [85]. Acute periods of NEB result in an increase in whole-body proteolysis, AA oxidation, and nitrogen excretion [85]. Interestingly, this proteolytic state in response to NEB becomes less pronounced and plateaus over an extended time period, presumably to enable the body to adapt and conserve protein and energy reserves [85,89]. To compound this enhancement in proteolysis, a human-based study revealed that a 10-day moderate NEB resulted in a 19% reduction in fasting SM protein synthesis [85,90]. (Other studies have shown discordant SM protein turnover data in response to NEB that may have resulted from differences in study design, populations, or dietary interventions [85].) Therefore, in response to sustained NEB, there is a downregulation of whole-body protein turnover including SM protein synthesis in the early stages, probably to conserve energy and protein reserves [85].

It is beyond the scope of this review to provide a detailed exposition of the intracellular regulation of the SM in response to NEB, which has been described eloquently elsewhere [85]. Here, we provide a summary. Intracellular signalling within myocytes of the SM respond to multiple factors that include energy status, nutrient availability, and growth factors [85]. The ‘Mammalian Target of Rapamycin Complex 1’ (mTORC1) is an important nutritionally regulated signalling component that influences mRNA translation and impacts on the insulin-signalling cascade and ultimately on protein synthesis within the SM [85]. ‘AMP-activated protein kinase’ (AMPK) also acts as a fuel sensor within the SM (with downstream inhibitory effects on mTORC1 activity [91]), and inhibits anabolic signalling pathways in response to limited energy availability and a reduction in cellular levels of ATP [85]. Based on the existing literature, the enhanced SM proteolysis that occurs within the SM is regulated through ‘caspase-mediated’ and ‘ubiquitin proteasome’ systems [85]. Given the role for insulin in the regulation of caspase 3 activity through phosphatidylinositol-3 kinase (PI3K), this provides a mechanism whereby enhanced SM proteolysis occurs in response to hypoinsulinaemia (a physiological feature of NEB) [85,90].

The inevitable loss of SM mass during NEB-induced body weight loss has important metabolic implications for dieting through the restriction of caloric intake. In particular, it has implications for ‘yo-yo dieting’, in which dieting through caloric restriction (with a consequent reduced body weight) is implemented recurrently over prolonged durations, with weight re-gain occurring within each ‘inter-dieting’ period (each cycle typically lasting many months) [92,93]. A problem with this fairly common and population-wide scenario is that during weight re-gain following initial weight loss, AT mass is re-gained to a greater degree than SM mass is [92,94]. To illustrate this, in one study on the body composition of postmenopausal women following intentional weight loss and subsequent body weight re-gain, for every 1 Kg of AT mass lost during the initial weight loss intervention, 0.26 Kg of lean tissue (primarily SM mass) was lost [92,94]. Conversely, for every 1 Kg of AT mass re-gained over the year following the initial weight loss, only 0.12 Kg of lean tissue was re-gained [92,94]. It follows that for every ‘yo-yo’ cycle of diet-induced weight loss and subsequent weight re-gain there is a 14% drop in overall lean mass, equating to an overall attrition of around 10% of SM mass with each ‘yo-yo’ dieting cycle. Such an attritional effect on SM mass would have adverse metabolic implications. Based on insights from this and other reported studies, it seems that from the perspective of SM mass and function (and overall metabolic health), a ‘floating’ body weight is preferable to one that ‘yo-yos’. In short, ‘slow and steady’ wins the metabolic race.

### 3.5. Preservation of SM Mass and Positioning in Future Management Strategies

We have discussed the importance of the SM for metabolic health and efficiency, and the vulnerability of the SM to a decline in its mass and functionality in response to aging, weight gain, and obesity, and to NEB as a common approach to effecting overall body weight loss in the context of obesity. In this section we explore strategies to protect our SM from the factors that serve to diminish it, to optimise its mass and functionality and thereby to preserve, maintain, and even improve metabolic health and efficiency. Broadly, there are two strategies: (i) lifestyle and (ii) novel pharmacotherapies.

#### 3.5.1. Lifestyle

One key lifestyle approach to mitigate against the diminishment of SM mass and function is a nutritional strategy to ensure the provision of a HPD: a diet that consists of a protein intake in excess of the usual recommended dietary allowance (0.8 g kg^−1^ day^−1^) [85]. There is evidence in the literature to support this approach. Interestingly, regarding the type of dietary protein there is no evidence that plant sources are any less suitable that animal sources [95]. Furthermore, it would seem sensible to avoid intermittent fasting given that several studies have demonstrated that this approach may increase the risk for loss of FFM and SM mass [96,97]. Indeed, the only diet that has been demonstrated to reduce cardiovascular risk regardless of age, sex, and T2D-status is the Mediterranean diet [98]. The Mediterranean diet also appears to have favourable effects on SM mass, including the attenuation of aging-related losses of lean mass even in the absence of a concomitant physical exercise programme [99,100].

Given the importance of myokines for the healthy functioning of the AMA and the release of myokines in response to physical exercise, it is logical that physical activity should also form a major component of the lifestyle approach for the preservation of SM mass and function. Resistance, aerobic, and combination training improves SM function and reduces body fat, with resistance training being particularly useful for improved SM performance [43]. The combination of resistance exercise training with dietary protein supplementation was assessed in a systematic review and meta-analysis, with significant improvements in SM strength and size [101]. The implementation of a HPD should ideally occur in combination with resistance physical exercise. Conversely, the implementation of a HPD without concurrent resistance physical exercise may actually worsen adiposity and overall IR and metabolic status [102,103]. However, there is some controversy in the literature regarding the necessity of resistance physical exercise concurrent with a HPD for the preservation of SM mass. Metabolic status at baseline is an important determinant of the metabolic response to a HPD without concurrent resistance physical exercise, with euglycaemia and T2D at baseline resulting in both worsening and improvement in metabolic status, respectively. The mechanism is such that for the SM to develop and utilise proteins in the diet, it requires simultaneous SM activity to provide a necessary stimulus (including muscle repair). Without this, there is no physiological mechanism to effectively utilise excessive dietary protein, which can then result in increased AT mass (depending on insulin sensitivity) through storage of these extra calories as fat, or putting additional strain on the kidneys [104]. Excessive dietary protein intake also stimulates urea synthesis, which requires energy and can help to induce NEB [105].

Regarding evidence to support the implementation of a HPD to mitigate against the loss of SM mass, this includes a study on postmenopausal women with obesity in which the extent of SM loss in response to NEB was proportional to dietary protein intake [85,106]. In this study, there was a lower reduction in FFM (17.3%) for those women consuming a HPD compared to those on a low-protein diet (FFM loss of 37.5%) [85,106]. Further studies have demonstrated the benefits of SM preservation through the consumption of a HPD in the context of prolonged periods of NEB [85,107,108]. In the context of NEB, negative nitrogen balance (including increased nitrogen excretion) generally occurs, and increasing dietary protein intake can offset this process [85]. Furthermore, the beneficial effects of a HPD on the preservation of SM mass in the context of NEB appear to occur regardless of whether the NEB is induced through dietary effects and/or physical activity. In one study in healthy young volunteers in which NEB was induced solely by increased aerobic-type physical activity over a 7-day period, doubling the dietary protein intake abrogated the increased nitrogen excretion and resultant negative nitrogen balance [85,109].

Regarding the type of dietary protein to mitigate against the loss of SM mass with NEB, branched-chain AAs (BCAAs) maintain SM protein synthesis and attenuate nitrogen excretion and muscle proteolysis [85,110]. Furthermore, our own group demonstrated that alterations in the complexity and signatures of AAs within the diet can have a major impact on whole-body and hepatic IR, and this process is likely mediated within the SM [111]. Amongst the BCAAs, leucine is a potent stimulator of SM protein synthesis, and in human studies the consumption of a leucine-containing food product during physical exercise decreased whole-body proteolysis and increased SM protein synthesis through stimulation of the mTORC1 pathway [85,112]. The amount of dietary leucine necessary to maximise the stimulation of anabolism within SM could be up to 7–12 g per day, which is much higher than the recommended daily intake of 14 mg kg^−1^ [85]. Whey protein has a high leucine content and its effects on SM strength and physical performance of older adults was reviewed recently [113]. Additional nutritional measures may include micronutrient supplements to the diet, including vitamin D, selenium, magnesium, and AAs to correct any pre-existing deficiencies [43].

#### 3.5.2. Novel Pharmacotherapies

Lifestyle approaches through HPD and resistance physical exercise can at least partially mitigate against the associated loss of SM mass in the context of NEB, as outlined. However, even with these measures in place, loss of SM mass seems inevitable (even without NEB, and just with the normal aging process). Furthermore, not everyone is able to engage fully in resistance physical exercise and/or comply with HPD. Therefore, there is a requirement for alternate strategies. The recent emergence of GLP1-based therapies that enable a body weight loss of >20% has promoted a current unmet clinical need for novel pharmacotherapies that effectively preserve and augment SM mass and therefore mitigate against the decline in SM mass in the context of such GLP1-based therapy-induced NEB. These therapeutic developments will have a broader clinical utility, including, for example, in the context of normal aging, obesity, and in concurrent chronic illness, all of which are associated with a diminution in SM mass.

In recent years, there has been a lot of interest in the therapeutic potential of myostatin inhibition to combat muscle atrophy [17]. Numerous molecules have been developed as potential myostatin antagonists, including follistatin and stamulumab (a myostatin antibody), although none have improved SM strength in muscle dystrophy patients and there have been some safety concerns [17]. Myostatin and Activin A are TGFβ-like ligands that signal via Activin type II Receptors (ActRII), and in the process antagonise SM growth [114]. Bimagrumab is a monoclonal antibody against ActRII, which could mitigate against the decline in SM mass in the context of GLP1-induced NEB. In a study on diet-induced obese mice, bimagrumab induced a reduction in fat mass with a 10% increase in lean mass [114]. In the same model, whilst daily treatment with semaglutide induced a significant reduction in both lean and fat mass, combination therapy with semaglutide and bimagrumab resulted in a superior loss of fat mass with a simultaneous preservation of lean mass (despite a reduction in food intake) and an improved metabolic status and exercise performance [114]. This study demonstrates proof of the concept, albeit in a rodent-based model, that combination therapy of a GLP1-based therapy with bimagrumab can improve body composition and protect the SM in the context of NEB. In a human-based study, bimagrumab was studied in adults with T2D and overweight and obesity in a 48 week, phase 2 randomised placebo-controlled clinical trial [115]. In the bimagrumab group, there was a significant reduction in fat mass (20.5%), an increase in lean mass (3.6%), a reduction in waist circumference (9 cm), and a reduction in HbA1C (0.76%) [115]. Bimagrumab was studied in a further human-based randomised placebo-controlled study in community-dwelling older adults with sarcopenia and mobility limitations [116]. In the bimagrumab group, there was a statistically significant increase in thigh muscle volume (7.72% increase at 16 weeks), and clinically meaningful improvements in gait speed for those participants with a slower walking speed at baseline [116].

Beyond the myostatin and Activin A pathway, the bioactive peptide Apelin (produced and secreted by the AT) provides an alternate potential therapeutic means of maintaining SM mass in the context of NEB. Azelaprag as a novel Apelin receptor (APJ) agonist has recently undergone early phase human-based studies, with pre-clinical data suggesting favourable effects on body composition and optimised SM mass and quality [117,118]. Furthermore, the endocannabinoid system may provide another therapeutic target for improving body composition and optimising SM mass. In a recently reported phase 1b study, the novel peripherally acting Cannabinoid Receptor-1 (CB1R) inverse agonist, INV-202 (monlunabant), was evaluated in a placebo-controlled trial over 28 days in participants with obesity and features of metabolic syndrome and glucose intolerance [119]. For those in the INV-202 group, there was a weight loss of 3.5 Kg and a significant reduction in waist circumference [119]. In addition to improved body composition, INV-202 may also have anti-fibrotic effects that are desirable for optimised kidney and liver functions in obesity and T2D [120]. In a recently reported rodent-based model, rimonabant—an older centrally acting CB1R inverse agonist—partially restored SM function and induced favourable metabolic changes in the AT and liver [121].

## 4. Discussion: AMR Versus BMI as a Diagnostic Criterion and Key Management Outcome for Obesity

As outlined earlier, obesity as a clinical entity is dominated by the BMI, including its diagnosis, severity, selection and response monitoring of management strategies, and even accessibility of key treatment options such as pharmacotherapies and bariatric surgery. This implicit international consensus for the BMI as a central tenet of obesity is unfortunate. The BMI is a blunt ratio that provides no indication of body composition (including location of the AT), can be misleading at extremes of body habitus, and lacks precision and accuracy regarding overall cardiometabolic risk. Obesity deserves and demands a more sophisticated assessment and definition, one that prioritises a management approach based on cardiometabolic risk, one that is fit for the 21st century, one that ultimately will facilitate and expediate a greater understanding of the quiddity of obesity, one that will develop and nurture a more compassionate approach, and one that will benefit people living with obesity. The development and implementation of reformed diagnostic criteria for obesity will require an international consensus that will take time. Furthermore, it is likely that novel techniques such as metabolomic assessments of serum and urine will feature more prominently in the diagnostics of the future, including for obesity. Just as with any scientific model, diagnostic criteria should be open to change and modification based on new insights into pathogenesis and techniques, and obesity should be no exception.

In this review, we have argued for the SM as a key determinant of overall metabolic functioning and efficiency, including the handling of macronutrients, IR, and BMR. The close anatomical and physiological links between the SM and AT enable it to function effectively as a single unit, the AMA. Given this novel insight based on the current literature, we argue that future revised definitions of obesity should include a measure of the AMA, to reflect an emphasis on overall body composition, and therefore cardiometabolic risk, which would in our view hold much greater clinical utility than the BMI. We propose the AMR as such a measure, representing a ratio of overall AT mass to overall SM mass. The AMA would overcome many of the shortages of the BMI, and it would be interesting to explore its predictive function vs. the BMI for mortality and incidence of chronic diseases.

Existing measures used in clinical practice provide insight into body composition, including, for example, the waist circumference and ‘Body–Adiposity Index’ (BAI: waist circumference/height), each of which provide a proxy measure of visceral fat content. Furthermore, the ‘Relative Body Fat Content’ (RBFC), measured through a variety of techniques, also provides useful insights into body composition from the perspective of AT content. Regarding the RBFC, there is controversy in the literature regarding its ability to predict the future risk of cardiovascular disease, with some negative studies [122,123,124,125,126], some positive studies [127,128], and some studies that are undecided regarding its clinical utility [129]. These measures of visceral and overall adiposity are clinically useful, and we argue that they provide greater insight into overall metabolic status than the BMI. However, neither of these measures factor in SM mass. Conversely, the AMR would factor in both SM and AT mass. In an extreme clinical scenario of SO, in which AT mass is elevated and SM mass is diminished, the AMR would be high, indicative of elevated cardiometabolic risk. Such a scenario should prompt a redoubling of management strategies to reduce the AMR, thereby improving cardiometabolic risk, and these types of cases should be prioritised. Conversely, in the example of extreme muscularity with relatively little AT, the AMR would be low (incidentally, the BMI would likely be high in such a scenario). A low AMR would not necessarily require any specific management approach. For all other cases between these two extreme clinical scenarios, the AMR would provide a useful measure of overall cardiometabolic risk on which to determine treatment prioritisation and individualised management strategies. As outlined, future treatment algorithms will include novel pharmacotherapies that target both the SM and AT, thereby directly improving the AMR.

For the AMR to have clinical utility, this will require the development of non-invasive and accurate measures of SM and AT mass, ideally performed at the point of care. The use of bioelectrical impedance techniques can provide these measures, although the accuracy and standardisation of this approach have been criticised [130]. Imaging techniques such as DXA and MRI can provide measures of body composition, although these are costly, take time to perform, and will never be scalable to a population level at which the AMR would be targeted. It is possible that in the future a myokine/adipokine signature from the serum or urine (redolent of urinalysis assessment) could provide a non-invasive, accurate, and rapid clinical assessment of SM and AT mass that in turn could provide the basis for a measure of AMR. In addition to its use for the diagnosis of obesity, the AMR would also provide an objective and clinically useful measure of treatment choice and response. This approach would transition emphasis away from the languid ‘percentage weight loss’ (stemming from the BMI), which dominates the obesity field currently (and has done for decades), and instead place the AMR centre stage, as a veracious and clinically meaningful reflector of cardiometabolic status.

## 5. Conclusions

Traditionally, obesity has suffered as a clinical entity. In addition to its originally intended use as an epidemiological tool to define populations, we place too much emphasis on the BMI to stratify and prioritise treatment options for individual people living with obesity, and even as a diagnostic criterion for obesity. With an emphasis on the BMI endorsed by key guideline-generating institutions such as NICE, we often ignore body composition in our clinical assessment of obesity, and in our treatment choices and monitoring of treatment responses. In our opinion, this over-reliance on the BMI has contributed towards key misconceptions regarding the quiddity of obesity. A simple ratio that implicates body weight and height reveals nothing of body composition, whereas the hardly more complicated BAI would provide insight into the visceral AT depot. However, a measure such as the AMR would provide an invaluable clinical biomarker of both the status and response to therapeutic interventions of the AMA, an arena in which the pathophysiology of obesity-related metabolic dysfunction plays out, and in which lies key therapeutic targets to optimise and improve metabolic health.

In this review, we have provided an overview of the multiple and complex links between the SM and AT, through complex endocrine and paracrine links that implicate multiple myokines and adipokines. We argue that rather than being two separate tissues, the SM and AT function as a single unit, the AMA. We have discussed the SM and its role in influencing overall metabolic health and BMR, and the vicious cycle of SO in its progression towards diminished SM mass and increased amounts of AT. We have discussed key lifestyle strategies to mitigate against the loss of SM mass in the context of NEB (a common approach to facilitate and induce weight loss), including optimised protein intake within the diet and engagement in resistance physical activities. We have discussed the active interest within the pharma and biotechnology industries to develop novel molecules that act within the periphery to maintain and even augment SM mass and diminish AT mass concurrently. In future, we will co-administer such molecules with GLP1-based therapies to mitigate against the loss of SM mass that typically occurs in response to the NEB induced by GLP1-based therapies. It is likely that such an approach will maintain and improve SM mass, which in turn will optimise overall metabolic functioning and efficiency and improve insulin sensitivity.

The SM seems to have suffered with regard to its role in obesity. One problem relates to current nomenclature and how this influences the human psyche. Due to its labelling as ‘fat’ by the layperson, there is an assumption that the AT is the main cause of obesity. The idea of the SM (in concert with the AT) being a key component of obesity pathogenesis and development is under-appreciated, and this misconception places too much emphasis on body weight reduction as a treatment target, rather than the optimisation of body composition and specifically the maintenance of SM mass and function. This over-emphasis on body weight reduction seems ubiquitous and even to have infiltrated the design of clinical trials for GLP1-related therapies. In our view, this focus on body weight reduction stems from the use of the BMI to define obesity and stratify its management approach. A shift in focus to developing and establishing the AMR as an alternative, easily obtainable measure reflecting the AMA would not only help to re-define obesity with regard to the SM and AT depots, but also promote an emphasis on body composition, and specifically the AMA as a treatment target in the context of obesity. This sea change in our perspectives is long overdue, and would, in our view, improve the lives of people living with obesity, and also help to de-stigmatise obesity and its origins through improved public understanding, which in turn would help to facilitate a more compassionate societal approach.

## Figures and Tables

**Figure 1 metabolites-15-00085-f001:**
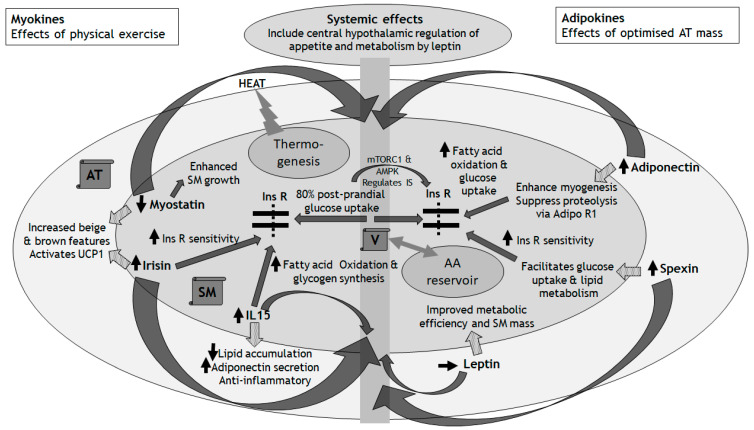
The Adipo–Muscle Axis (AMA). Dotted arrows show paracrine effects of myokines and adipokines on the AT and SM, respectively; solid curved arrows show systemic endocrine effects via the vasculature; and solid straight arrows show effects on Ins R sensitivity, glucose, and amino acid handling within the SM. AA: amino acids; Adipo R1: receptor for adiponectin; AMPK: AMP-activated protein kinase; AT: adipose tissue; IL-15: interleukin-15; Ins R: insulin receptor; IS: insulin signalling; mTORC1: Mammalian Target of Rapamycin Complex 1; SM: skeletal muscle; UCP1: uncoupling protein 1; V: vasculature.

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
