# Peer review of "Optimised Skeletal Muscle Mass as a Key Strategy for Obesity Management"

_metabolites, 2025, doi:10.3390/metabo15020085_

Round 1
Reviewer 1 Report
Comments and Suggestions for Authors
The work by Barber et al., Optimised Skeletal Muscle Mass as a Key Strategy for Obesity Management, provides a narrative review of the role played by skeletal muscle in metabolism, mainly associated with adipose tissue, and its interpretation for the most effective diagnosis of obesity and future treatments. Although there are studies for the diagnosis of obesity that consider clinical diagnostic markers in addition to the body mass index, the work is interesting because the prevention and management of obesity is regarded as a global problem. It is necessary to improve the tools for personalized treatment.
Some observations for improving the work are proposed below:
1. In the Abstract, it is not very clear what the objective of the work is and how the topic will be addressed to achieve it; the authors mention what information they reviewed, but what was their objective?
2. In the introduction, you could mention those diseases that are associated with the highest number of deaths and obesity, for example cancer; also, note what are the consequences of obesity and overweight on the quality of life, even for aging; as well as the importance of establishing adequate measures for the prevention, diagnosis and treatment of obesity since it is now considered a public health problem worldwide, and that even the WHO classifies it as a non-infectious pandemic.
3. Also mention that there are metabolic markers of obesity subtypes, where in addition to BMI they consider fat levels, muscle volume, good fitness, insulin sensitivity, blood glucose levels, and cardiovascular risk. (https://doi.org/10.1111/dme.15226)
4. What are the limitations of using only the BMI to establish the nutritional status of a person? Or in any case, to diagnose obesity?
5. In the methodology it would be important to consider in the keywords, subtypes or phenotypes of obesity. How do they measure perceived clinical relevance? Did they include all types of articles? For example, meta-analysis, short communication, reviews, letters to the editor, clinical cases??
6. In the results, could it be considered in “3.1 The physiological role of the SM, and its decline with age”, to include aging, the above due to the fact that “Age is a linear concept that is expressed in years, while aging is a multifactorial process that is determined by genetic and environmental factors and that begins from birth”. Also, Skeletal muscle aging is a key contributor to age-related frailty and sarcopenia with substantial implications for global health. Could it be relevant to include the types of sarcopenia involved? could consider the following references:
Rodríguez-Rodero S, Fernández-Morera JL, Menéndez-Torre E, Calvanese V, Fernández AF, Fraga MF. Aging genetics and aging. Aging Dis. 2011 Jun;2(3):186-95. Epub 2011 Apr 28. PMID: 22396873; PMCID: PMC3295054.
Shen, X., Wang, C., Zhou, X., Zhou, W., Hornburg, D., Wu, S., & Snyder, M. P. (2024). Nonlinear dynamics of multi-omics profiles during human aging. Nature aging, 4(11), 1619–1634. https://doi.org/10.1038/s43587-024-00692-2
Reyes-Farias, M., Fos-Domenech, J., Serra, D., Herrero, L., & Sánchez-Infantes, D. (2021). White adipose tissue dysfunction in obesity and aging. Biochemical pharmacology, 192, 114723. https://doi.org/10.1016/j.bcp.2021.114723
Kedlian, V. R., Wang, Y., Liu, T., Chen, X., Bolt, L., Tudor, C., Shen, Z., Fasouli, E. S., Prigmore, E., Kleshchevnikov, V., Pett, J. P., Li, T., Lawrence, J. E. G., Perera, S., Prete, M., Huang, N., Guo, Q., Zeng, X., Yang, L., PolaÅ„ski, K., … Zhang, H. (2024). Human skeletal muscle aging atlas. Nature aging, 4(5), 727–744. https://doi.org/10.1038/s43587-024-00613-3
7. For point 3.2, you could consider the following references:
An SM, Cho SH, Yoon JC. Adipose Tissue and Metabolic Health. Diabetes Metab J. 2023 Sep;47(5):595-611. doi: 10.4093/dmj.2023.0011. Epub 2023 Jul 24. PMID: 37482656; PMCID: PMC10555533.
Rocha-Rodrigues, S., Matos, A., Afonso, J., Mendes-Ferreira, M., Abade, E., Teixeira, E., Silva, B., Murawska-Ciałowicz, E., Oliveira, M. J., & Ribeiro, R. (2021). Skeletal Muscle-Adipose Tissue-Tumor Axis: Molecular Mechanisms Linking Exercise Training in Prostate Cancer. International journal of molecular sciences, 22(9), 4469. https://doi.org/10.3390/ijms22094469
Shen, S., Liao, Q., Zhang, T., Pan, R., & Lin, L. (2019). Myricanol modulates skeletal muscle-adipose tissue crosstalk to alleviate high-fat diet-induced obesity and insulin resistance. British journal of pharmacology, 176(20), 3983–4001. https://doi.org/10.1111/bph.14802
8. In point 3.2.1 they could include that Musclin, LIF, IL-4, IL-6, IL-7 and IL-15 promote muscle hypertrophy and myostatin inhibits muscle hypertrophy. (Severinsen MCK, Pedersen BK. Muscle-organ crosstalk: The Emerging Roles of Myokines. Endocr Rev. 2020 Aug 1;41(4):594–609. doi: 10.1210/endrev/bnaa016. Erratum in: Endocr Rev. 2021 Jan 28;42(1):97-99. doi: 10.1210/endrev/bnaa024. PMID: 32393961; PMCID: PMC7288608.)
9. In the figure, add the AMPK, mTORC, PI3K1 pathways, among others; how are they regulated or involved?
Author Response
Reviewer 1
Comment 1.1:
The work by Barber et al., Optimised Skeletal Muscle Mass as a Key Strategy for Obesity Management, provides a narrative review of the role played by skeletal muscle in metabolism, mainly associated with adipose tissue, and its interpretation for the most effective diagnosis of obesity and future treatments. Although there are studies for the diagnosis of obesity that consider clinical diagnostic markers in addition to the body mass index, the work is interesting because the prevention and management of obesity is regarded as a global problem. It is necessary to improve the tools for personalized treatment. Some observations for improving the work are proposed below:
In the Abstract, it is not very clear what the objective of the work is and how the topic will be addressed to achieve it; the authors mention what information they reviewed, but what was their objective?
Response to Comment 1.1:
Thank you for this comment. We agree with the reviewer. Accordingly, to clarify the overall objective of our review, we have included some additional text in the abstract of the revised version.
Comment 1.2:
In the introduction, you could mention those diseases that are associated with the highest number of deaths and obesity, for example cancer; also, note what are the consequences of obesity and overweight on the quality of life, even for aging; as well as the importance of establishing adequate measures for the prevention, diagnosis and treatment of obesity since it is now considered a public health problem worldwide, and that even the WHO classifies it as a non-infectious pandemic.
Response to Comment 1.2:
Thank you for this comment. We agree with the reviewer that these are important points. Accordingly, we have included some additional text to the introduction section of the revised version of our manuscript.
Comment 1.3:
Also mention that there are metabolic markers of obesity subtypes, where in addition to BMI they consider fat levels, muscle volume, good fitness, insulin sensitivity, blood glucose levels, and cardiovascular risk. (https://doi.org/10.1111/dme.15226)
Response to Comment 1.3:
Thank you for this comment. We agree with the reviewer that this is an important point. Accordingly, we have included some additional text and the suggested reference into the introduction section of the revised version of our manuscript.
Comment 1.4:
What are the limitations of using only the BMI to establish the nutritional status of a person? Or in any case, to diagnose obesity?
Response to Comment 1.4:
Thank you for this comment. With respect, we already provided a detailed discussion of the limitations of BMI for use as a diagnostic criterion for obesity in paragraph 3 of the introduction in the original version of our manuscript. These problems and limitations include lack of any indication of overall adiposity or body fat distribution (each of which are key criteria that should contribute towards any clinical assessment of obesity). The limitations of BMI in the clinical assessment of obesity, as outlined, pertain especially at extremes of body habitus (with excessive or severely diminished muscularity). Regarding the nutritional status of a person, this is a complex clinical question that requires a detailed nutritional history, clinical assessment and both serum and urine biochemical assays to fully appreciate. Crudely, nutritional status is reflected by body composition (including degree of adiposity and muscularity). However, given that BMI fails as a useful measure of body composition, we can conclude therefore that BMI also has limitations to assess nutritional status. To address the failings of BMI as a useful clinical indicator of both obesity and nutritional status, we have added some additional text to the introduction section of the revised version of our manuscript.
Comment 1.5:
In the methodology it would be important to consider in the keywords, subtypes or phenotypes of obesity. How do they measure perceived clinical relevance? Did they include all types of articles? For example, meta-analysis, short communication, reviews, letters to the editor, clinical cases??
Response to Comment 1.5:
We are grateful for this comment. We agree with the reviewer that these are important considerations. We did not include any subtypes or phenotypes of obesity in the search terms. We included all types of peer-reviewed articles appearing on the Pubmed search, including narrative reviews, systematic reviews and meta-analyses, short communications, clinical cases and original articles. To clarify these points, we have included some additional text in the methodology section of the revised version of our manuscript.
Comment 1.6:
In the results, could it be considered in “3.1 The physiological role of the SM, and its decline with age”, to include aging, the above due to the fact that “Age is a linear concept that is expressed in years, while aging is a multifactorial process that is determined by genetic and environmental factors and that begins from birth”. Also, Skeletal muscle aging is a key contributor to age-related frailty and sarcopenia with substantial implications for global health. Could it be relevant to include the types of sarcopenia involved? could consider the following references:
Rodríguez-Rodero S, Fernández-Morera JL, Menéndez-Torre E, Calvanese V, Fernández AF, Fraga MF. Aging genetics and aging. Aging Dis. 2011 Jun;2(3):186-95. Epub 2011 Apr 28. PMID: 22396873; PMCID: PMC3295054.
Shen, X., Wang, C., Zhou, X., Zhou, W., Hornburg, D., Wu, S., & Snyder, M. P. (2024). Nonlinear dynamics of multi-omics profiles during human aging. Nature aging, 4(11), 1619–1634. https://doi.org/10.1038/s43587-024-00692-2
Reyes-Farias, M., Fos-Domenech, J., Serra, D., Herrero, L., & Sánchez-Infantes, D. (2021). White adipose tissue dysfunction in obesity and aging. Biochemical pharmacology, 192, 114723. https://doi.org/10.1016/j.bcp.2021.114723
Kedlian, V. R., Wang, Y., Liu, T., Chen, X., Bolt, L., Tudor, C., Shen, Z., Fasouli, E. S., Prigmore, E., Kleshchevnikov, V., Pett, J. P., Li, T., Lawrence, J. E. G., Perera, S., Prete, M., Huang, N., Guo, Q., Zeng, X., Yang, L., PolaÅ„ski, K., … Zhang, H. (2024). Human skeletal muscle aging atlas. Nature aging, 4(5), 727–744. https://doi.org/10.1038/s43587-024-00613-3
Response to Comment 1.6:
Thank you for this comment. We agree with the reviewer that ‘aging’ is a better term than ‘age’ for the reasons given. Accordingly, we have included the term ‘aging’ in section 3.1 of the revised version of our manuscript. As suggested, we have also included some additional text and references in this revised section. Regarding an overview of the different types of sarcopenia, we did include discussion of both aging-related and obesity-related sarcopenia in the original version of our manuscript.
Comment 1.7:
For point 3.2, you could consider the following references:
An SM, Cho SH, Yoon JC. Adipose Tissue and Metabolic Health. Diabetes Metab J. 2023 Sep;47(5):595-611. doi: 10.4093/dmj.2023.0011. Epub 2023 Jul 24. PMID: 37482656; PMCID: PMC10555533.
Rocha-Rodrigues, S., Matos, A., Afonso, J., Mendes-Ferreira, M., Abade, E., Teixeira, E., Silva, B., Murawska-Ciałowicz, E., Oliveira, M. J., & Ribeiro, R. (2021). Skeletal Muscle-Adipose Tissue-Tumor Axis: Molecular Mechanisms Linking Exercise Training in Prostate Cancer. International journal of molecular sciences, 22(9), 4469. https://doi.org/10.3390/ijms22094469
Shen, S., Liao, Q., Zhang, T., Pan, R., & Lin, L. (2019). Myricanol modulates skeletal muscle-adipose tissue crosstalk to alleviate high-fat diet-induced obesity and insulin resistance. British journal of pharmacology, 176(20), 3983–4001. https://doi.org/10.1111/bph.14802
Response to Comment 1.7:
Thank you for this comment. We agree with the reviewer that these are useful references to include. Accordingly, we have added these references to section 3.2 in the revised version of our manuscript.
Comment 1.8:
In point 3.2.1 they could include that Musclin, LIF, IL-4, IL-6, IL-7 and IL-15 promote muscle hypertrophy and myostatin inhibits muscle hypertrophy. (Severinsen MCK, Pedersen BK. Muscle-organ crosstalk: The Emerging Roles of Myokines. Endocr Rev. 2020 Aug 1;41(4):594–609. doi: 10.1210/endrev/bnaa016. Erratum in: Endocr Rev. 2021 Jan 28;42(1):97-99. doi: 10.1210/endrev/bnaa024. PMID: 32393961; PMCID: PMC7288608.)
Response to Comment 1.8:
Thank you for this comment. We agree with the reviewer. In response, we have added some additional text and the additional suggested references to section 3.2.1 of the revised version of our manuscript.
Comment 1.9:
In the figure, add the AMPK, mTORC, PI3K1 pathways, among others; how are they regulated or involved?
Response to Comment 1.9:
Thank you for this comment. We agree with the reviewer that these are important considerations. We already provided an overview of mTORC1 and AMPK and their roles in the regulation of insulin signalling and protein synthesis within the skeletal muscle in section 3.4 of the original version of our manuscript. Within the figure, there is a limit to how much detail can be provided to avoid cluttering and to preserve readability. To address this comment, we have included an additional arrow within the revised figure, connecting glucose uptake into the skeletal muscle with the insulin signalling pathway (via the insulin receptor), this pathway being mediated by changes in mTORC1 and AMPK as fuel sensors.
Reviewer 2 Report
Comments and Suggestions for Authors
Once my comments are implemented, the paper will be suitable for publication.

Author Response
Reviewer 2
Comment 2.1:
Title: Optimised Skeletal Muscle Mass as a Key Strategy for Obesity Management Line 45: In the introduction, you provide a strong background on the limitations of BMI, SM, and AT in evaluating obesity. However, I recommend touching on Active Body Mass (ABM) as a potential evaluation tool (a more meaningful and practical alternative to the current measures). Moreover, I encourage the authors to explore ABM in more detail, primarily in the discussion section.
Response to Comment 2.1:
We thank the reviewer for this interesting suggestion. We agree that there are many ways to assess body composition and physical fitness. Furthermore, we acknowledge the important and relevance of physical fitness to overall metabolic health. However, given the concise nature of our review and the focus on skeletal muscle and its metabolic sequelae, we feel that a broader discussion of the clinical measures of body composition and physical fitness (beyond that which is already provided in our original manuscript) are out of the scope of our review. We have done a thorough Pubmed search on the term ‘Active Body Mass’ and were unable to identify any existing scientific and peer-reviewed papers on this subject. We did a Google search of the same term which also revealed no relevant published papers on this topic. Furthermore, the ‘Active Body Mass’ index is not something that the authors are familiar with. Therefore, we are unable to add any additional text or references referring to or discussing ‘Active Body Mass’ in a meaningful way, and which would add value and utility to our review. We respectfully appreciate the reviewer’s suggestion for inclusion and discussion of this term but are unable to properly address this for the reasons outlined.
Comment 2.2:
Line 130: I recommend mentioning sarcopenia, cachexia, and atrophy together.
Response to Comment 2.2:
Thank you for this comment. In response, we have included some additional text in the introduction section of our revised manuscript to confirm that sarcopenia is a form of SM atrophy. We have not included any mention of cachexia, as this is beyond the scope of our concise review which focuses on sarcopenia in the context of obesity, and its metabolic implications.
Comment 2.3:
Line 143: I highly recommend including a chart of the reviewed papers. It will provide transparency, clarity, and an organized summary of the search process and the selected studies.
Response to Comment 2.3:
Thank you for this comment. Please see also our response to comment 1.5. In the methods section of the revised version of our manuscript, we have included some additional text to provide further clarity of the search and selection process, with inclusion of all types of peer-reviewed articles. We included details of the search terms for Pubmed in the original version of our manuscript. Our review is fully referenced, with a list of all cited references included already.
Comment 2.4:
Line 150: The authors address a broad range of interconnected but distinct topics, which can dilute the paper's focus. The paper should be better structured with clearly defined subsections to streamline the results and discussion section and ensure clarity. From my perspective, the authors should focus more on BMI, SM, AT, ABM, AMR, sarcopenia, cachexia, and atrophy and their diagnostic tools, emphasizing the importance of shifting toward AMB/AMR as a more meaningful metric or concentrate on AMA and metabolic health and Sarcopenia and metabolic health…Avoid overloading, and narrow the focus by emphasizing the paper's primary aim – whether it is a critique of current metrics, an argument for adopting ABM/AMR, or an exploration of sarcopenic obesity. This will ensure a more transparent narrative.
Response to Comment 2.4:
Thank you for this comment. In our original manuscript, we already provide a focus on BMI, SM, AT, AMR, sarcopenia and sarcopenic obesity, including diagnostic approaches and we already emphasize the importance of shifting toward AMR as a more clinically meaningful biomarker of metabolic health compared with BMI. Clearly labelled subsections are provided. As was mentioned in our response to comment 2.1, we are unable to include any details on ‘Active Body Mass’ in our revision. Furthermore, we feel that a broader discussion of cachexia would enter the realm of malignancy-related effects on the skeletal muscle (which has been reviewed comprehensively and in detail elsewhere) and would therefore detract from the main emphasis of our approach, which is to focus on skeletal muscle in the context of metabolic control and efficiency and obesity. With respect, we do not feel that we have ‘overloaded’ the reader with excessive detail and our approach throughout is to provide a concise review. We do accept, however, that the topic for discussion is broad and complex. To properly address the reviewer’s comment would involve re-writing the entire manuscript and the deletion of much of the existing content which we feel would ultimately deprive the readership of Metabolites from gaining important and timely insights on this topic. With respect, therefore, we have chosen not to re-structure our entire manuscript in the way suggested but thank the reviewer again for their comment.